# Study of Long-Term Determination Accuracy for REEs in Geological Samples by Inductively Coupled Plasma Quadrupole Mass Spectrometry

**DOI:** 10.3390/molecules26020290

**Published:** 2021-01-08

**Authors:** Xijuan Tan, Minwu Liu, Ke He

**Affiliations:** 1Laboratory of Mineralization and Dynamics, Chang’an University, 126 Yanta Road, Xi’an 710054, China; minwul@chd.edu.cn (M.L.); zyheke@chd.edu.cn (K.H.); 2College of Earth Sciences and Land Resources, Chang’an University, 126 Yanta Road, Xi’an 710054, China

**Keywords:** REE determination accuracy, long-term stability, ICP-QMS, geological samples

## Abstract

This work presents the long-term determination accuracy study of ICP-QMS for rare earth elements (REEs) in geological matrices. Following high-pressure closed acidic decomposition, REEs are measured repetitively across seven months by ICP-QMS. Under optimum experimental conditions (including spray chamber temperature, gas flow rate, sampling depth, etc.), the REE contents in geological standard materials from basic (basalt BCR-2 and BE-N) to intermediate (andesite AGV-2) and up to acidic (granite GSR-1) show good agreement with the certified values, giving relative errors below 10%. Here, the influence of two storage materials (perfluoroalkoxy PFA and polypropylene PP) on the long-term determination accuracy of REEs has also been monitored. It is found that the relative errors of REEs using a PFA container range from −6.6 to 6.3% (RSDs < 6.0%), while that using a PP container are within −4.0 to 3.9% (RSDs < 4.6%). By using PP material as a solution storage container, the accuracy of REEs quantification in a series of real geological samples are checked, showing the RSDs of less than 5.0%. This work first clarifies the long-term stability of REEs quantification by ICP-QMS covering two types of storage materials, confirming the reasonability of PP material as a daily storage container in terms of higher data precision and lower cost.

## 1. Introduction

Rare earth elements (REEs), which are proven to exist in a wide range of geological settings, consist of the lanthanide elements from La (Z = 57) to Lu (Z = 71) [1]. Generally, REEs are dominantly trivalent in terrestrial rocks and show decreasing ionic radii from 1.06 Å for La^3+^ to 0.85 Å for Lu^3+^ [2]. Due to similar physical and chemical properties, the REEs seem to be resistant to fractionation in supracrustal environments and immobile in most metamorphic conditions [3]. Coupled with low solubility and immobility in the terrestrial crust, the REEs distribution pattern, such as anomalies in La, Ce, Eu and Gd, and the enrichment or depletion of light-REEs and heavy-REEs, makes this group valuable in both basic and applied geological studies [4,5,6]. 

To apply the REEs distribution pattern in the investigation of the associated geological activities, the accurate determination of REEs in geological samples is required. However, the chemical and physical similarities among REEs cause difficulties and complications in the quantification of REEs in geological samples. Furthermore, analytical problems become considerable, especially when assaying the selected REE with low concentrations [7]. Consequently, the development of accurate analytical methods for REEs determination has steadily received greater attention during the last several decades [8,9,10]. 

Besides energy-dispersive X-ray analysis (EDXA) [11], X-ray fluorescence analysis (XRF) [12,13] and electrothermal atomic absorption spectrometry (ETAAS) [14], the most widely used determination methods for REEs appear to be neutron activation analysis (NAS) [15,16], inductively coupled plasma optical emission spectrometry (ICP-OES) [17,18] and ICP-mass spectrometry (ICP-MS) [19,20]. Although NAA is a sensitive technique that only needs small amounts of samples for analysis, this method suffers from time-consuming, requiring a neutron generation reactor and serious matrix interferences, which limit its extensive application in routine assays of REEs [9]. ICP-OES is more convenient for rapid REEs determination, but the detection limits are not low enough, making complicated separation and preconcentration necessary [8]. ICP-MS is a well-established analytical method, which is characterized by small sample volume and low detection limits, extremely high sensitivity, wide dynamic range, simple spectra and multi-element analytical capability [21,22]. Currently, ICP-MS is the most attractive technique for REEs determination [23]. However, the accuracy and stability for REEs quantification by ICP-MS remain challenging due to the strict control of the blank level and the notorious REEs oxide ion interferences [24,25]. Ardini et al. [26] reported the successful direct quantification of REEs in glaciomarine sediments by ICP-OES, ICP-quadrupole MS (ICP-QMS) and ICP-sector field MS (ICP-SFMS), in which reliable REEs data with high accuracy and good precision, however, can be obtained using ICP-OES by carefully selecting the emission lines and doing an internal standardization to compensate for non-spectral interferences. Kasar et al. [27] investigated the digestion efficiencies of microwave and Savillex decomposing techniques for geological certified reference materials in the application of soil REEs determination by ICP-MS/MS, and they showed that the former digestion approach was more effective and faster. Satyanarayanan et al. [28] evaluated the analytical capability of high-resolution (HR)-ICP-MS for REEs in geological standard materials, arguing that the uncertainty value is far superior to that of ICP-QMS due to the higher m/z and less interference of HR-ICP-MS. Whitty-Lé veillé et al. [29] also compared the quantification accuracy and precision of ICP-MS/MS to other plasma-based instruments, including microwave-induced plasma-OES (MIP-OES), ICP-OES and ICP-QMS for REEs in various certified mineral matrices, showing ICP-MS/MS coupled with alkaline fusion digestion was the better choice for assaying variable levels of REEs. For MS-based analytical tools, the quantification accuracies of REEs by ICP-SFMS, HR-ICP-MS and ICP-MS/MS are higher than that of ICP-QMS, while the corresponding assay cost derived from ICP-QMS is relatively lower. Although ICP-QMS is a sophisticated technique in REEs measurement, no relevant study has been focused on the long-term quantification stability of ICP-QMS for REEs in geological samples up to date.

It is known that the blank level is a crucial factor in trace and ultra-trace element analysis [30]. In general, the blank level of a given sample is affected by the purity of sample-treating reagents, ambient analytical environment and container materials. Accordingly, great efforts and considerations have been placed on blank value reduction, such as controlling an ambient analytical environment in a clean room, enhancing the reagent purity by sub-boiling distillation and thoroughly cleaning utilized labware [31,32]. Since possible contaminations might be introduced from containers used during sample treatment and storage, bottles fabricated from fluorocarbon polymers (including polytetrafluoroethylene PTFE, fluorinated ethylenepropylene FEP and perfluoroalkoxy PFA) have become popular alternatives to polyethylene (PE) or polypropylene (PP) [33]. Despite having reportedly clean, non-stick and reusable properties, these specific polymer containers are extremely expensive and inconvenient for large-batch laboratory analysis, which raises the question “whether it is an absolute necessity to use such material”. To the best of our knowledge, there is no clear viewpoint regarding the effects of storage materials on the long-term determination accuracy for REEs by ICP-QMS. 

The first aim of this study is to trace the long-term quantification accuracy of ICP-QMS for REEs. Here, geological standard materials including BCR-2 (basalt), BE-N (basalt), AGV-2 (andesite) and GSR-1 (granite) were repeatedly taken for REEs measurement over the course of seven months. The second aim of this study is to clarify the effect of PFA and PP bottles as solution storage containers on the quantification accuracy of REEs by ICP-QMS. The REEs results of the standard geological materials stored in PFA and PP bottles were analyzed by relative errors (REs) and relative standard deviations (RSDs) in detail. This study can provide valuable information to answer the above proposed question and clarify the long-term assay capability of ICP-QMS for REEs in geological samples.

## 2. Results and Discussion

### 2.1. Optimization of Spray Chamber Temperature

Severe polyatomic interferences exist in the direct determination of REEs, such as the formation of oxide and hydroxide ions of light REEs (La, Ce, Pr and Nd), which greatly affect the determination accuracy of heavy REEs (Gd, Tb and Dy), and the formation of BaO and BaOH lead to spectral overlaps on both ^151^Eu and ^153^Eu [34]. Compared to standard spray chamber systems at 20 °C, a cooling spray chamber at 0 °C was reported to effectively minimize oxide formation: the percentage ratios of REEs oxide formation were lower than 5% in a 10% HNO_3_ (*v*/*v*) solution [26]. 

In this work, we also examined the effect of the spray chamber temperature on the REEs oxide formation in a 2% HNO_3_ (*v*/*v*) solution Because of the risk of condensation in the spray chamber caused by the freezing of the 2% HNO_3_ (*v*/*v*) solution, which exhibits a freezing temperature of about −0.6 °C, the tested spray chamber temperatures range from 2 to 22 °C. The results showed that the oxide ratios of REEs decline when the spray chamber temperature decreases. Therefore, a spray chamber temperature of 2 °C was selected as the optimal one. At this temperature, the formation of oxide ions were observed to drop nearly 60% compared to that produced when the spray chamber temperature was set at 22 °C. It was also found that the interference errors can be reduced with the decrement in formed oxide ions, giving an interference error of less than 4.8% for a 10 ng/mL of REEs standard solution at 2 °C of spray chamber. Thus, a spray chamber temperature of 2 °C was used during the subsequent study.

### 2.2. Operating Parameter Optimization of ICP-QMS 

For ICP-QMS analysis, daily optimization of operating parameters is mandatary. With the spray chamber temperature set at 2 °C, the effects of Ar gas flow rates (including nebulizer gas flow, plasma gas flow and auxiliary gas flow), nebulizer inserting depth, sampling depth and sampling flow are fully studied. 

To optimize nebulizer gas flow, its influences on the ratios of oxide formation, hydroxide formation and doubly charged species were tested. Taking Ce as the representative element, the results are shown in Figure 1. It can be seen that the ratios of CeO^+^/Ce^+^ and CeOH^+^/Ce^+^ slightly increase with an increase in the nebulizer gas flow from 0.6 to 0.8 L/min, with corresponding values well under 3.0%, and then ascend sharply. However, the ratio of Ce^2+^/Ce^+^ first goes up to 0.75 L/min, showing a maximum value of 3.3%, and then decreases to a value lower than 2.5% under 0.85 L/min of nebulizer gas flow. Additionally, the corresponding signal intensity of Ce shows an obvious increment in the range from 0.6 to 0.85 L/min and then an apparent decrement from 0.85 to 1.2 L/min. Thus, 0.85 L/min of nebulizer gas flow is recommended for the subsequent testing. Here, the effect study for plasma gas flow rate and auxiliary gas flow rate showed that the optimum values were 14.5 L/min and 0.8 L/min, respectively. To further enhance the sensitivity and precision of REEs quantification, the possible effect of nebulizer inserting depth in the spray chamber was also investigated. Results revealed that the signal intensities of REEs were greatly influenced by the nebulizer inserting depth, showing that REEs signals sharply increased with a decrease in the nebulizer inserting depth from 36 to 30 mm, and then declined slightly. Thus, a nebulizer inserting depth of 30 mm was chosen as the optimal depth. After taking signal stability and efficiency of reagents into consideration, the other ICP-QMS operating parameters, 90 of sampling depth and 1.0 mL/min of peristatic pump, were selected in the subsequent experiments.

In order to test the long-term instrumental operation stability of ICP-QMS, the effect of the chosen daily gas flow rates (0.85 L/min of nebulizer gas flow, 14.5 L/min of plasma gas flow and 0.8 L/min of auxiliary gas flow) on the ratios of Ce^2+^/Ce^+^, CeO^+^/Ce^+^ and CeOH^+^/Ce^+^ and Ce signal intensity for 10 ng/mL of REEs standard solution were also monitored over seven months. Results showed that Ce signal intensity was relatively stable around the level of 4.2 × 10^4^ CPS, and all the ratio values of Ce^2+^/Ce^+^, CeO^+^/Ce^+^ and CeOH^+^/Ce^+^ were less than 3.5% across the whole assay period, with inter RSDs well below 6.5% (*n* = 50), confirming the reasonability of the daily operating parameters of this ICP-QMS instrument for REEs quantification. 

### 2.3. REEs Determination Results for Sample Solution Stored in PFA Material

Under the optimum experimental conditions, the REEs in a series of geological reference materials including basic basalts (BCR-2 and BE-N), intermediate andesite (AGV-2) and acidic granite (GSR-1) were then analyzed over the course of seven months. By using Rh as the online internal standard element, the digested sample solutions kept in PFA bottles were determined repeatedly by ICP-QMS. The obtained REEs concentrations and the 95% confidential intervals (2σ) for 25 individual analyses of each sample were summarized in Table 1. Here, all the listed REEs results were the average values of five repetitive measurements for each parallel study, with inner determination RSDs (*n* = 5) less than 1.0% and the inter determination RSDs in the range from 1.0% to 5.9%. It is clear in Table 1 that the REEs of the four studied standard materials agree with the certified values, confirming the assay stability of the ICP-QMS under the chosen operating parameters and no significant effect of PFA storage material on REEs quantification accuracy. 

### 2.4. REEs Determination Results for Sample Solution Stored in PP Material 

Considering the PP bottle is the routine storage container for digested sample solution in our laboratory, only GSR-1 was applied to study the effect of PP material as the storage material on REEs quantification accuracy. The concentration results with the corresponding 95% confidential intervals (2σ, *n* = 25) listed in Table 1 show that the REEs of GSR-1 are in agreement with the referred values, reconfirming the operation stability of the ICP-QMS under the daily instrumental parameters. Here, the inter-determination RSDs for REEs in GSR-1 are in the range of 1.5% to 2.9%, which demonstrates there is negligent influence of PP storage material on quantification accuracy of REEs in geological samples.

In addition to the geological standard material GSR-1, another five real geological samples, with the digested solutions stored in PP bottles, were also taken for long-term quantification accuracy of REEs by ICP-QMS. The obtained REEs results with 95% confidential intervals (2σ, *n* = 25) are summarized in Appendix A. There are no significant variations of REEs concentrations among the five repetitive measurements across seven months, indicating that PP storage material doesn’t affect the quantification accuracy and precision.

### 2.5. Long-Term Stability Assessment of REEs Quantification Covering Storage Materials

From the above results, the obtained REEs of the four geological standard materials agree with the certified values no matter which type of storage material is used. Further REs analysis for REEs in BCR-2, BE-N, AGV-2 and GSR-1, with digested solution stored in PFA bottles, show that the RE values are from −6.6 to 6.3% (see Figure 2a–d). However, the RE values for GSR-1 with digested solution stored in a PP bottle are from −4.0 to 3.9% (see in Figure 2e). Therefore, it can be deduced that sample solution stored in a PP bottle give REEs values with slightly higher precision than that kept in a PFA bottle. Furthermore, the inter-determination RSDs for REEs in the real geological samples in Figure 3 clearly reveal that the RSDs over seven months are less than 5.0%, which reconfirms the PP bottle type is a reliable storage material for REEs quantification in geological study. In addition, considering the market price of the two storage types (i.e., roughly 760 RMB per 50 mL of PFA bottle and around 0.67 RMB per 50 mL of PP bottle), the PP bottle type is highly recommended for solution storage in large sample batch assays. Although the PFA bottle is reusable, the strict clean procedures prior to usage are quite complicated, which might introduce potential contaminants and add too much work. It can be concluded that PP material is a better choice than PFA material as the storage container in routine laboratory analysis.

## 3. Materials and Methods

### 3.1. Reagents and Chemicals

High-purity acids and ultra-pure water with a resistivity of 18.2 MΩ·cm were used throughout analysis. Before usage, the commercially available acids including HNO_3_ (68% *v*/*v*, AR grade) and HF (40% *v*/*v*, AR grade) were purified twice using sub-boiling distillation in Teflon stills (Savillex DST-1000-PFA, Savillex Corporation, Eden Prairie, MN, USA) to remove metallic or cationic residues. Deionized water passed through a Milli-Q water purification system (Millipore, Bedford, MA, USA) to produce high-purity water.

Five solutions (5, 10, 20, 50 and 100 ng/mL for all the elements) in 2% HNO_3_ (*v*/*v*), which were used as the external calibrators, were prepared progressively by gravimetric dilution method from 10 μg/mL of Multi-element Calibration Standard solutions (Agilent Technologies, Tokyo, Japan). Mono-element solution was prepared from 1.0 mg/mL of single element standard solution, which was purchased from the National Institute of Standards and Technology, China. Here, to exclude any possible assay bias from long-term storage, all standard solutions were prepared fresh. The used PFA bottles and pipet tips were first immersed in 50% HNO_3_ (*v*/*v*) for 12 h and then heated in purified water at 50 °C for 8 h. Prior to standard solution preparation, the PFA bottles and pipet tips were carefully rinsed three times using high-purity water.

### 3.2. Instrumental Apparatus

The instrument is a Thermofisher Scientific X series ICP-QMS (Waltham, MA, USA). This ICP-QMS apparatus is equipped with a concentric nebulizer, a cyclonic spray chamber wrapped by a cone chamber with impact bead, a standard quartz torch, an assemble of Ni sample/skimmer cones (1.1/0.75 mm), a quadrupole mass analyzer and a peristaltic pump.

The ICP-QMS instrument worked under 1250 W of forward power and was optimized daily to obtain stable and relative maximum intensities for Ce and U using a 10 ng/mL of tuning solution. Meanwhile, the ratios for oxide formation (CeO^+^/Ce^+^), hydroxyl formation (CeOH^+^/Ce^+^) and doubly charged species (Ce^2+^/Ce^+^) were controlled under 3.0%. Thereafter, a rock solution was introduced to flush the system for at least 30 min to minimize instrumental drift. During the measurement, a standard rock digested solution as the drift monitor was repeatedly analyzed for every five unknown samples. The memory-related background from the previous sample was resolved by continuously washing the system in a 2% HNO_3_ (*v*/*v*) solution, with a count level of 10 ng/mL of Rh checked. The data were read under peak jumping mode with channel spacing of 0.02 and a dwell time of 10 ms.

### 3.3. Geological Standard and Sample Materials

Four commonly used rock reference materials in a geological laboratory, including basic BCR-2 (basalt) and BE-N (basalt), intermediate AGV-2 (andesite) and acidic GSR-1 (granite), were selected in the long-term stability study of REEs quantification covering PFA and PP storage materials in this work. The BCR-2 and AGV-2 are USGS geochemical reference materials (USA), BE-N is CRPG standard material (France), and GSR-1 is IGGE standard material (China). To further evaluate the effect of storage material on REEs determination accuracy, a series of real rock samples finely grounded less than 200 mesh was also analyzed.

### 3.4. Sample Decomposition Procedure

All digestion Teflon bombs were cleaned in aqua regia at 120 °C for 12 h and then transferred into high-purity water for another 12 h at 120 °C. Prior to usage, the labware was carefully rinsed three times with high-purity water. The samples including reference materials and real rock samples were digested in a class 1000 clean room following a high-pressure closed acidic decomposition method developed by Tan et al. [35]. In brief, samples with a quantity of 50 ± 0.5 mg were weighed in 15 mL of Teflon bombs, and 1.0 mL of HF and 0.5 mL of HNO_3_ were added into the samples gently. After the acids had mixed with samples, the bombs were placed on the hotplate and evaporated to incipient dryness at 140 °C. Thereafter, 1.0 mL of HF and 1.0 mL of HNO_3_ were added into the samples, and the bombs were sealed in high-pressure metal jackets before being transferred into an oven at 185 °C for 12 h. After cooling, the bombs were carefully opened and put on a hotplate at 140 °C. When becoming incipiently dry, the samples were fortified with 1.0 mL of HNO_3_ and again evaporated to incipient dryness. With 2.5 mL of 40% HNO_3_ (*v*/*v*) added, the residues were re-dissolved at 135 °C for 6 h with bombs in metal jackets and then aged overnight. The final solutions were transferred to PFA or PP bottles and then gravimetrically diluted to 50 ± 0.5 mg using a 2% HNO_3_ (*v*/*v*) solution. Finally, the sample solutions were taken for REEs quantification by ICP-QMS directly.

## 4. Conclusions

This work fully investigated the long-term quantification accuracy of ICP-QMS for REEs in geological matrices covering two types of storage materials for the first time. Results showed that the contents of REEs in four geological standard materials (basic basalt BCR-2 and BE-N, intermediate andesite AGV-2 and acidic granite GSR-1) agreed with the certified values over the course of seven months. However, compared to storage material made of PFA with REs from −6.6 to 6.3%, storage material made of PP exhibited relatively lower REs from−4.0 to 3.9%. The use of PP bottles as storage containers for a series of real geological samples also convinced the reliability of this storage material, with inter determination RSDs of less than 5.0% in REEs measurement. Clearly, this study provides valuable information to obtain accurate concentrations of REEs in geological samples by ICP-QMS. It is also now clear that it is not an absolute necessity to use storage containers made from fluorocarbon polymer material, and that PP material is highly recommended and offers advantages, including long-term stability of REEs quantification, lower cost and less labor.

## Figures and Tables

**Figure 1 molecules-26-00290-f001:**
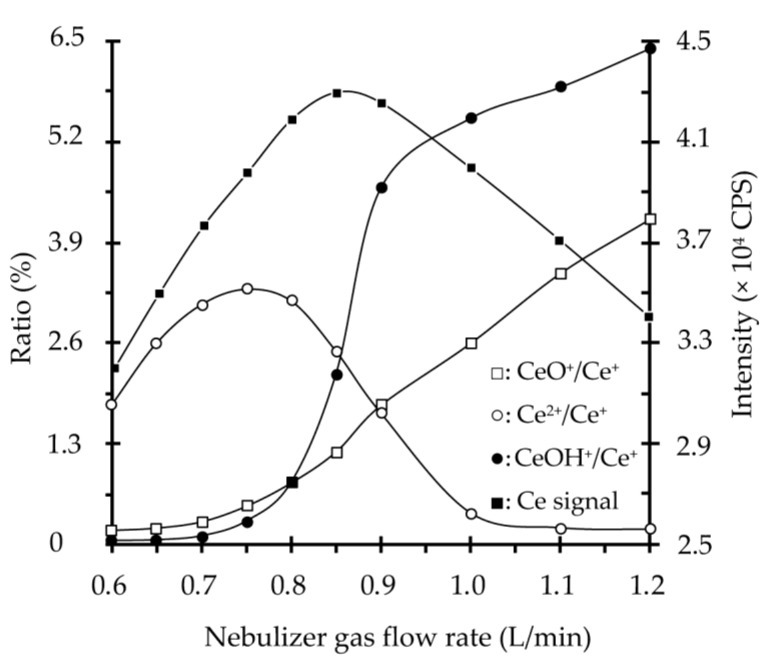
The effect of nebulizer gas flow rate on ICP-QMS analyzing accuracy. With an increasing nebulizer gas flow rate of 0.6–1.2 L/min, (□): the ratio trend of CeO^+^/Ce^+^; (o): the ratio trend of Ce^2+^/Ce^+^; (●): the ratio trend of CeOH^+^/Ce^+^; and (■): the signal trend of Ce.

**Figure 2 molecules-26-00290-f002:**
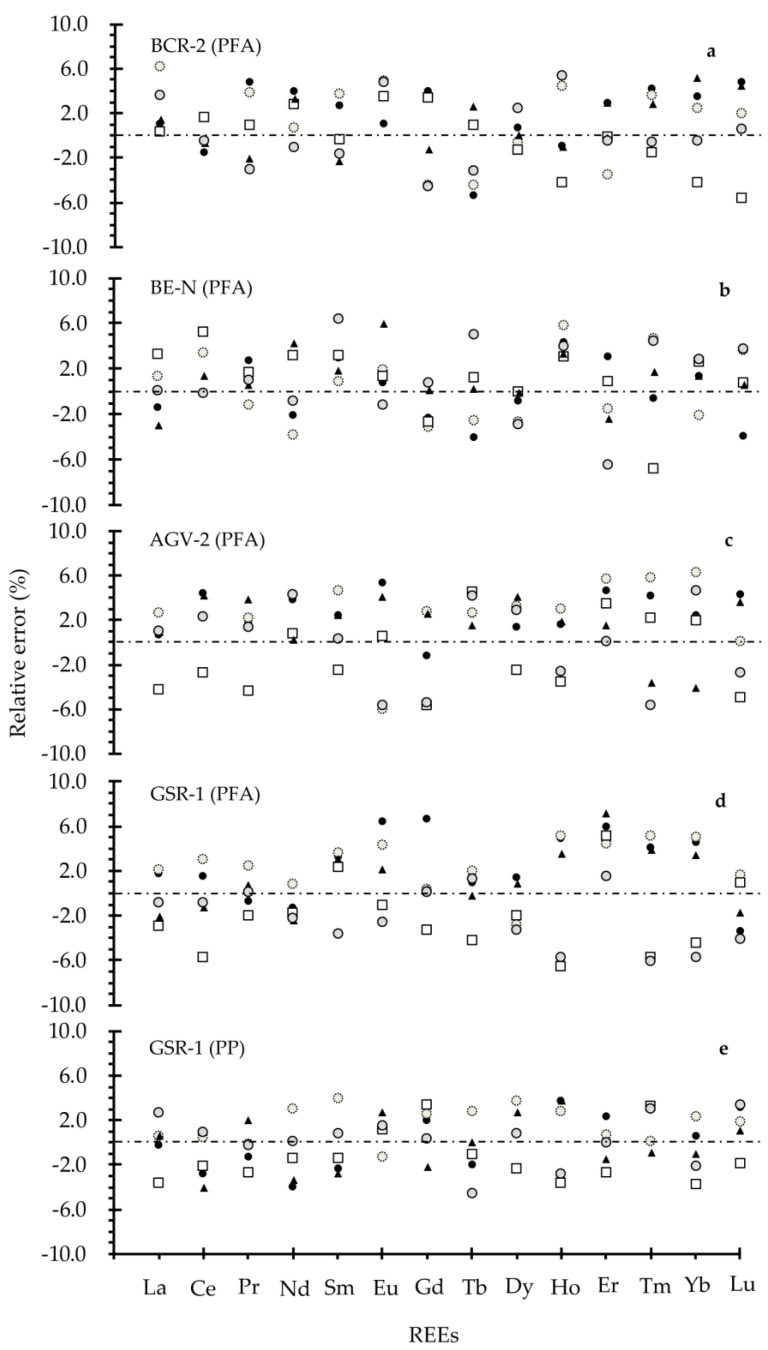
The RE values of REEs in geological standard materials with different storage bottle types. (**a**–**d**): a PFA bottle is used as the storage container for digested solutions of BCR-2, BE-N, AGV-2 and GSR-1, respectively; (**e**): a PP bottle is used as the storage container for digested solutions of GSR-1.

**Figure 3 molecules-26-00290-f003:**
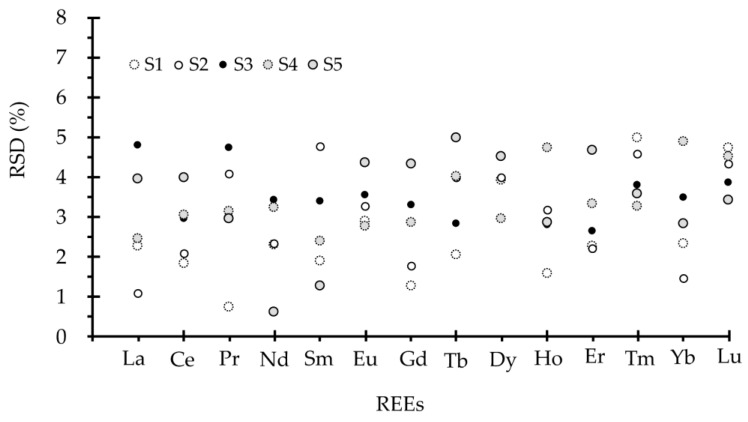
The RSDs of REEs determination in real rock samples using PP bottles as storage containers. The RSDs of REEs for real rock samples are calculated from the repetitive determination results within seven months.

**Table 1 molecules-26-00290-t001:** REEs of geological standard materials with PFA and PP storage containers.

BCR-2		La	Ce	Pr	Nd	Sm	Eu	Gd	Tb	Dy	Ho	Er	Tm	Yb	Lu
Referred value		25.0	53.0	6.80	28.0	6.70	2.00	6.80	1.07	6.41	1.33	3.66	0.54	3.50	0.51
PFA	1 day	25.3	52.2	7.13	29.1	6.88	2.02	7.07	1.01	6.45	1.32	3.77	0.56	3.62	0.53
	7 days	26.5	53.8	7.06	28.2	6.94	2.10	6.49	1.02	6.37	1.39	3.53	0.56	3.58	0.52
	30 days	25.1	53.9	6.86	28.8	6.67	2.07	7.03	1.08	6.33	1.27	3.65	0.53	3.35	0.48
	6 months	25.4	52.6	6.66	28.9	6.54	2.10	6.72	1.10	6.41	1.32	3.77	0.56	3.68	0.53
	7 months	25.9	52.7	6.59	27.7	6.59	2.10	6.48	1.04	6.56	1.40	3.64	0.54	3.48	0.51
	2σ	0.24	0.30	0.09	0.23	0.07	0.01	0.11	0.01	0.04	0.02	0.04	0.01	0.05	0.01
BE-N		La	Ce	Pr	Nd	Sm	Eu	Gd	Tb	Dy	Ho	Er	Tm	Yb	Lu
Referred value		82.0	152.0	17.5	67.0	12.2	3.60	9.70	1.30	6.40	1.10	2.50	0.34	1.80	0.24
PFA	1 day	80.8	159.8	18.0	65.6	12.6	3.62	9.47	1.25	6.35	1.15	2.57	0.34	1.82	0.23
	7 days	83.1	157.2	17.3	64.4	12.3	3.67	9.39	1.27	6.23	1.16	2.46	0.36	1.76	0.25
	30 days	84.6	159.9	17. 8	69.1	12.6	3.65	9.44	1.32	6.40	1.13	2.52	0.32	1.85	0.24
	6 months	79.5	154.0	17.6	69.9	12.4	3.81	9.71	1.30	6.39	1.14	2.44	0.35	1.83	0.24
	7 months	82.0	151.7	17.7	66.4	13.0	3.55	9.78	1.36	6.21	1.14	2.34	0.35	1.85	0.25
	2σ	0.79	1.44	0.10	0.93	0.10	0.04	0.07	0.02	0.04	0.005	0.04	0.01	0.01	0.003
AGV-2		La	Ce	Pr	Nd	Sm	Eu	Gd	Tb	Dy	Ho	Er	Tm	Yb	Lu
Referred value		38.0	68.0	8.30	30.0	5.70	1.54	4.69	0.64	3.60	0.71	1.79	0.26	1.60	0.25
PFA	1 day	38.2	71.0	8.45	31.1	5.84	1.62	4.63	0.66	3.65	0.72	1.87	0.27	1.64	0.26
	7 days	39.0	66.2	8.48	30.3	5.97	1.45	4.82	0.66	3.72	0.73	1.89	0.28	1.70	0.25
	30 days	36.4	66.1	7.94	30.2	5.56	1.55	4.43	0.67	3.51	0.69	1.85	0.27	1.63	0.24
	6 months	38.4	70.9	8.63	30.1	5.85	1.60	4.81	0.65	3.75	0.72	1.82	0.25	1.54	0.26
	7 months	38.4	69.6	8.42	31.3	5.72	1.45	4.44	0.67	3.71	0.69	1.79	0.25	1.67	0.24
	2σ	0.40	0.98	0.10	0.23	0.06	0.03	0.08	0.003	0.04	0.01	0.02	0.01	0.03	0.004
GSR-1		La	Ce	Pr	Nd	Sm	Eu	Gd	Tb	Dy	Ho	Er	Tm	Yb	Lu
Referred value		54.0	108	12.7	47.0	9.70	0.85	9.30	1.65	10.2	2.05	6.50	1.06	7.40	1.15
PFA	1 day	54.9	110	12.6	46.4	10.0	0.90	9.91	1.67	10.3	2.15	6.89	1.10	7.73	1.11
	7 days	55.1	111	13.0	47.3	10.0	0.89	9.33	1.68	9.90	2.15	6.78	1.11	7.76	1.17
	30 days	52.4	102	12.5	46.1	9.92	0.84	8.99	1.58	9.99	1.91	6.83	1.00	7.07	1.16
	6 months	52.9	107	12.8	45.9	9.99	0.87	9.34	1.65	10.3	2.12	6.96	1.10	7.65	1.13
	7 months	53.5	107	12.7	45.9	9.34	0.83	9.31	1.67	9.86	1.93	6.60	0.99	6.98	1.10
	2σ	0.48	1.44	0.08	0.24	0.12	0.01	0.13	0.02	0.09	0.05	0.05	0.02	0.15	0.01
PP	1 day	53.9	105	12.5	45.1	9.48	0.86	9.49	1.62	10.3	2.13	6.65	1.10	7.44	1.19
	7 days	54.3	108	12.7	48.4	10.1	0.84	9.54	1.70	10.6	2.11	6.54	1.06	7.57	1.17
	30 days	52.1	106	12.4	46.3	9.56	0.86	9.61	1.63	9.96	1.98	6.33	1.10	7.12	1.13
	6 months	54.4	104	13.0	45.4	9.43	0.87	9.09	1.65	10.5	2.13	6.41	1.05	7.33	1.16
	7 months	55.5	109	12.7	47.1	9.78	0.86	9.34	1.57	10.3	1.99	6.50	1.09	7.25	1.19
	2σ	0.50	0.93	0.09	0.54	0.11	0.01	0.08	0.02	0.10	0.03	0.05	0.01	0.07	0.01

## Data Availability

The data presented in this study are available in Appendix A.

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
