# Peer review of "Study of Long-Term Determination Accuracy for REEs in Geological Samples by Inductively Coupled Plasma Quadrupole Mass Spectrometry"

_molecules, 2021, doi:10.3390/molecules26020290_

Round 1
Reviewer 1 Report
Review of a ms entitled “Study of Long-term Determination Accuracy for REEs in Geological Samples by Inductively Coupled Plasma Quadrupole Mass Spectrometry” submitted to Molecules as molecules-1031295 R1
As it stands the objective of the present ms is not very clear and it should be emphasized. Hence the authors examine the stability of REE (rare earth elements) after 7 months of storage within perfluoroalkoxy PFA and polypropylene PP material. The errors are below 6 %, using ICP-QMS.
The introduction is a long paragraph stating the aim of measuring REE in earth sciences, using different techniques from ICP-MS to ICP-QMS. Probably it relates to more modern techniques, more sophisticated (and less expensive). This is not stated. I understand that the authors want to better “sell” their method, but this is not explicit for the usual reader not enough involved in REE measurements.
Apparently, the error stands below a good value (6%) , but this should be compared to the standard error on the measurements in a single repeated shot.
A last comment relates to the issue in Molecules, journal that may have few diffusion in Earth Sciences to which it is destinated.
In conclusion, the present could be accepted with Minor modifications
Author Response
Query 1. The introduction is a long paragraph stating the aim of measuring REE in earth sciences, using different techniques from ICP-MS to ICP-QMS. Probably it relates to more modern techniques, more sophisticated (and less expensive). This is not stated. I understand that the authors want to better “sell” their method, but this is not explicit for the usual reader not enough involved in REE measurements.
Response: To make the paragraph stating REEs quantification techniques in earth sciences more clearly, the analytical properties of different methods especially MS-based techniques including ICP-SFMS, HR-ICP-MS, ICP-MS/MS and ICP-QMS for REEs determination were added in the Introduction section.
Query 2. Apparently, the error stands below a good value (6%), but this should be compared to the standard error on the measurements in a single repeated shot.
Response: Yes, in a single repeated measurement for REEs in the geological standard materials and real samples, the errors were calculated and expressed in RSDs. Results showed that the inner determination RSDs for all the REEs were well below 1.0%, which confirmed the daily working stability of the utilized ICP-QMS. Still, the errors within seven months were shown as 2σ form and listed in Table 1 and Table S1.
Reviewer 2 Report
The manuscript entitled “Study of Long-term Determination Accuracy for REEs in Geological Samples by Inductively Coupled Plasma Quadrupole Mass Spectrometry” describes the long-term determination accuracy of determination of rare earth elements by ICP-QMS with the with distinction to different types of solution storage materials. Unfortunately, the experiment seems to lack a novelty in terms of the analytical procedure, sample preparation and the metrological approach and in the current form I do not recommend it to be published in the Molecules. The described experiments seems similar to the experiments performed by the CRM manufacturers, namely the long-term storage stability of analytes, which is usually described in detail in certification report. I would recommend the authors to focus more on the different approaches of solid sample preparation for the analysis by ICP-QMS, and provide more information on chemical metrology, which would enrichen the results and discussion part of the manuscript. The Results and Discussion chapter, without tables, takes a little above one page. The discussion is too concise and should be expanded.
The manuscript requires a thorough check from the native english speaker.
Below are some minor remarks (verse):
- 20 – the word “stuff” does not sound right in the scientific paper in my opinion; rather use “material”, “packaging” or “container”;
- 35 heavy-REY – should be REE;
- 47 and others - ICP-AES is a historical acronym, the proper one that should be used is ICP-OES; AES is auger electrons spectroscopy;
- 57, 67 and many more – It is not advisable to write contraction or short forms of English words, like it’s for it is or aren’t for are not;
- 75 - polyethene (PE) – the chemical name “polyethylene” is advisable;
- 88 - The quotation marks "" are redundant here in my opinion;
- 188 - 50% HNO3 for cleaning seems to be excessively high acid concentration;
- 217 – The authors probably meant class 1000 clean room;
references 34 – Wrong DOI nr – it leads to a different article; I suggest to check other positions in the references as well.
Author Response
Query 1. I would recommend the authors to focus more on the different approaches of solid sample preparation for the analysis by ICP-QMS, and provide more information on chemical metrology, which would enrichen the results and discussion part of the manuscript. The Results and Discussion chapter, without tables, takes a little above one page. The discussion is too concise and should be expanded.
Response: Because the details of sample preparation method was presented in our previous work (Tan, X.J., et al., J. Anal. Chem. 2020, 75, 1295. Doi: 10.1134/S1061934820100147), here we just described the digestion procedures in brief in the section of 3.4 Sample Decomposition Procedure.
We expanded the Results and Discussion section according to the reviewer’s suggestion, (1) the section of 2.1. Optimization of Spray Chamber Temperature was rewritten, (2) the description of the effect of nebulizer inserting depth on signal sensitivity was supplemented in the section of 2.2. Operating Parameter Optimization of ICP-QMS, (3) the section of 2.3. REEs Determination Results for Sample Solution Stored in PFA Material was rewritten.
Query 2. The manuscript requires a thorough check from the native english speaker.
Response: The spellings were examined, the grammars were carefully scrutinized, and the sentences of the whole manuscript were polished.
Query 3. Below are some minor remarks (verse):
- 20 – the word “stuff” does not sound right in the scientific paper in my opinion; rather use “material”, “packaging” or “container”;
Response: The word “stuff” was changed to container in the whole manuscript.
- 35 heavy-REY – should be REE;
Response: Yes, the heavy-REY should be heavy-REEs, and the correct form was given in the text.
- 47 and others - ICP-AES is a historical acronym, the proper one that should be used is ICP-OES; AES is auger electrons spectroscopy;
Response: All the ICP-AES was changed to ICP-OES.
- 57, 67 and many more – It is not advisable to write contraction or short forms of English words, like it’s for it is or aren’t for are not;
Response: The short forms including “it’s” and “aren’t” were changed to “it is” and “are not” in the whole manuscript.
- 75 - polyethene (PE) – the chemical name “polyethylene” is advisable;
Response: The original polyethene (PE) was changed to “polyethylene”.
- 88 - The quotation marks "" are redundant here in my opinion;
Response: The quotation of “whether it is an absolute necessity to use such material” was deleted according to reviewer’s suggestion.
- 188 - 50% HNO3 for cleaning seems to be excessively high acid concentration;
Response: Actually, there was an acid cleaning step after the 12 hr-immersion of PFA bottles and pipet tips in 50% HNO3, and the corresponding description was added in the section of 3.1 Reagents and Chemicals.
- 217 – The authors probably meant class 1000 clean room;
Response: Yes, the reviewer is right. It is a class 1000 clean room and it was clarified in the section of 3.4 Sample Decomposition Procedure.
Query 4. references 34 – Wrong DOI nr – it leads to a different article; I suggest to check other positions in the references as well.
Response: Yes, we made a mistake. The correct information for reference 34 was given. It should be as follows.
May, T.W.; Wiedmeyer, R.H. A table of polyatomic interferences in ICP-MS. At. Spectrosc. 1998, 19, 150–155. Available online: https://www.researchgate.net/publication/298889298.
Reviewer 3 Report
See attached file.
The English language of the manuscript would benefit from a check form a native English speaker, if the authors have one among their colleagues.

Author Response
Query 1. Line 20 (and may other places): Do not use ‘stuff’, use material or container or something more describing.
Response: The word “stuff” was changed to container in the whole manuscript.
Query 2. Table 1: It is common to report uncertainties as (2σ), the 95% confidence interval. Here only the 1σ, 67% confidence interval is reported. The reported uncertainties should preferably be changed to 2σ, or at the very least it should be specified that they are 1σ.
Response: Thanks for this valuable suggestion. In Table 1 and Table S1, the 95% confidential intervals (2σ) of obtained REEs concentrations of the geological standard materials and real samples were added, and the inter RSD values were deleted.
Round 2
Reviewer 2 Report
The authors have improved the quality of the manscript so it is suitable to publish in Molecules. However, some minor errors and typos are still present in text and should be corrected, perhaps in the proof prior to the publishing.
Examples of errors are below, but authors should again thoroughly check the entire manuscript and supplementary material:
verse 48 - inductively coupled plasma ...
v. 153 - 4.2×104 153 CPS
v. 188 - could not
suppl. v. 17 - container
Author Response
Query 1. verse 48 - inductively coupled plasma ...
Response: The “inductively coupled optical emission spectrometry” was changed to “inductively coupled plasma optical emission spectrometry”.
Query 2. v. 153 - 4.2×104 153 CPS
Response: The “4.2×104 ICPS” was changed to “4.2×104 CPS”.
Query 3. v. 188 - could not
Response: The “couldn’t” was changed to “could not”.
Query 4. suppl. v. 17 - container
Response: “The “contianer” was changed to “container” in the supplemental material.